# New Insight into the Genome-Wide Diversity and Admixture of Six Colombian Sheep Populations

**DOI:** 10.3390/genes13081415

**Published:** 2022-08-09

**Authors:** Herman Alberto Revelo, Vincenzo Landi, Diana López-Alvarez, Yineth Alexandra Palacios, Samuel R. Paiva, Concepta McManus, Elena Ciani, Luz Ángela Alvarez

**Affiliations:** 1Departamento de Ciencia Animal, Facultad de Ciencias Agropecuarias, Universidad Nacional de Colombia, Palmira 763533, Colombia; 2Facultad de Medicina Veterinaria y Zootecnia, Universidad San Martin Cali Colombia, Carrera 122 #23-395 del, Vía Cali-Puerto Tejada, Cali 760022, Colombia; 3Dipartimento di Medicina Veterinaria, University of Barii “Aldo Moro”, 70010 Valenzano, BA, Italy; 4Departamento de Ciencias Biológicas, Facultad de Ciencias Agropecuarias, Universidad Nacional de Colombia, Palmira 763533, Colombia; 5Embrapa Recursos Geneticos e Biotecnologia, Brasılia 70770-917, Distrito Federal, Brazil; 6Instituto de Biologia, Universidade de Brasılia, Brasılia 70910-900, Distrito Federal, Brazil; 7Dipartimento di Bioscienze, Biotecnologie e Biofarmaceutica, University of Bari “Aldo Moro”, 70010 Bari, BA, Italy

**Keywords:** animal genetic resources, genomics, admixture, SNPs, Ovis Aries, South America, Africa, differentiation

## Abstract

Creole sheep represent a strategic genetic resource for populations living in marginal areas under financial restrictions on the American continent. Six Colombian sheep breeds (two wool (BCL-Boyacá and NCL-Nariño, 12 and 14 samples) and four hair (OPCE-Ethiopian, 54 samples; OPCS-Sudan, 74 samples; OPCP-Pelibeuy, 59 samples; OPCW-Wayúu, 24 samples) were genotyped using the Illumina Ovine SNP50 BeadChip. Data was also included from international 44 breeds from International Sheep Genomics Consortium (ISGC) and from data published in previous a previous work on the Caribbean and African breeds. Although geographically separated, wool (NCL, BCL) and hair types (OPCE, OPCS, OPCW) presented little genetic differentiation (FST 0.05) at a global level but several groups of animals separated suggesting local clustering due to geographical isolation. The OPCP underwent a recent crossing with Mexican Pelibuey, explaining its differentiation. Findings in this work such as the proximity to West African Djallonké (WAD) and Barbados Black Belly (BBB), suggest different introductions of African type animals from the Caribbean region on a pre-existing genetic basis formed by animals deriving from the first importations coming from Europe in colonial times. As expected, Colombian wool breeds showed, in particular in Admixture software results, a greater genomic component in common with European breeds and in particular with Iberian ones (Churra). This study provides a basis for future research into the genetic diversity within and between the Colombian sheep breeds analysed, and scientific data for policy decisions on Farm Animal Genetic Resources (FAnGR).

## 1. Introduction

In the American continent, Creole sheep represent a strategic genetic resource for populations living in marginal areas with scarce economic resources [1,2,3]. They also represent an essential role in the family economy since they provide high-quality meat, wool and, leather to local populations [4,5,6]. The term creole (criollo in Spanish or crioula in Portuguese) refers to the Iberian descendants in the Americas, including cattle and sheep [2]. Phenotypic comparisons of the wool characteristics [7] and mitochondrial DNA [8] demonstrated that sheep breeds in old Spanish-Latin American colonies were closely related to the Spanish Manchega, Latxa, and Churra breeds. Merino-type animals are unlikely to have been introduced during colonization due to the strict protection laws of the Castile Kingdom [9]. Still, various Merino-derived types of origin have been introduced in more recent times, mainly in the Andean regions [10]. West African hair sheep were probably introduced using the same route of the slave trade when the sugar cane industry started in the region [0]. The African genetic types were widely crossed with those already established in the Americas, presuming that the latter had acquired specific traits due to natural selection, including resistance to parasites, prolificacy, adaptation to hostile environments, food shortages, and extreme temperatures [11]. There is no consensus on the arrival of hair sheep of African origin, which may have reached the Caribbean at different times, compared to wool sheep that arrived from the Iberian Peninsula [2]. Further introduction of exotic breeds (Suffolk, Hampshire, Lacaune) could have occurred since the 1960s and 1980s, although there are no official records/reports [5]. In recent years, synthetic breeds, such as the Dorper and Katadhin or the Mexican Pelibuey, were also introduced, leading to the progressive replacement of the original population [2,8,12]. As a result, the dynamics of the Colombian sheep evolution process are unclear, and the reconstruction of their populations’ history and genetic structure is complex [2,5,13]. Wool sheep types are raised mainly in the Andean and southwestern regions of Colombia, while in the tropical regions, hair sheep types are most common on small subsistence farms [2]. In an earlier study on mitochondrial DNA, we found the prevalence of European haplotype despite the hypothesis of Central African origin of some of these populations [2]. In Colombia, local sheep breeds represent the 95.8% of the total sheep census and are mainly reared in the region of La Guajira (43.6%), Magdalena (10.8%), César (9.3%), Boyacá (8.3%), Córdoba (4.0%), Santander (2.6%), Bolívar (2.6%), Cundi-namarca (2.6%), Meta (2.3%), Sucre (1.9%), Valle del Cauca (7.8%) (https://www.ica.gov.co/, accessed on 26 March 2021). Previous studies have described several populations using morphology, microsatellites, and mitochondrial DNA markers [2,5,14]. Among them, the Etiope or sometimes called Abissino (Ethiopian or Abyssinian), Sudan (Sudanese), are raised mainly in the tropical region of the Caribbean (Córdoba, Cesar, Valledupar, Madalena) under traditional production systems in the absence of pedigree control and reproductive technologies. In the Valle del Cauca region (Pacific Coast, Colombian southwest), the most common reared population is the Pelibuey (literally ox-like hair), used along with sugar cane crops and are used for weed control [2,5]. his term also identifies a large and heterogeneous group of animals in all tropical parts of the American continent. On the Guajira peninsula, the Wuayu hair sheep are adapted to arid environments, high temperatures, low rainfall (200–400 mm), and drought, the production systems are developed in horizontal transhumance [2]. With the Lanado Creole, Colombian people generally identify wooled sheep type raised in the highlands. Two populations have been identified in the Andean region of Boyacá and Nariño [2,5]. The objective of this study was to expand the knowledge about the population structure of Colombian Creole sheep and their relationships at the worldwide level using genome-wide SNP data.

## 2. Materials and Methods

### 2.1. Sampling and DNA Extraction

A total of 237 samples were collected from the six breeds previously described [2]. The number of samples and flock collected per breed and geographical region are in Table 1. In Revelo et al. [2] a photo of typical individuals of each breed as well as the geographical position of sampling can be consulted.

Animals were ~50% of both sexes, excluding OPCP, where all samples were from males. A total of 10 mL of blood from each animal were collected in BD vacuum tubes (Becton Dickinson, NJ, USA) with k3-EDTA. Genomic DNA was extracted from 237 blood samples using the QIAamp^®^ DNA Mini Kit (QIAGEN, Hilden, Germany). DNA quantity and quality were evaluated by spectrophotometry using Colibri^TM^ (Titertek Berthold, Germany) and agarose gel electrophoresis using a standard protocol to evaluate sample degradation. Additionally, six samples of Ecuadorian Creole sheep were included as representatives from the nearest country in the region.1. Genotyping and dataset preparation.

The genotyping was performed using an Illumina Ovine SNP50 BeadChip Infinium^®^ Assay, including 54,241 SNPs (Weatherbys Scientific Ltd., Naas, Co. Kildare, Ireland). Using the PLINK v1.9b5.2 program [15], the raw data were merged with publicly available data to select ad-hoc worldwide sheep breeds for comparative statistical analysis. Data from the international HapMap sheep project [16] represented European, African and International breeds. Spangler et al. [1] were used to retrieve Caribbean populations and the Djallonke breed representative of West African genetic types. Finally, data from the study published by Paim et al. [3] permitted comparison with other creole breeds in the nearest geographical region. After removing unmapped and sexual chromosome variants, a total of 46216 SNPs was retained. Again using the PLINK v1.9b5.2 program [15], data were filtered using the following parameters: call-rate per marker, < 95% (2230 variants filtered out); MAF, < 0.01 (9104 variants filtered out). The final global dataset consisted of 34,882 SNPs and 1535 samples. The final dataset composition is reported in Appendix A.

Separately the process with the same parameters was repeated for only the Colombian population resulting in a dataset of 42,119 SNPs and 237 samples (4097 variants removed due to missing genotype only).

### 2.2. Statistical Analysis

#### 2.2.1. Genetic Diversity

Observed and expected heterozygosity (H_o_ and H_e_) was estimated for each breed using the *‘--het*’ and *‘--family*’ commands in PLINK [15]. The pairwise fixation index (FST) was calculated as a measure of genetic diversity between populations [17] using the snpgdsFst function of the ‘*SNPRelate*’ R package [18] and visualised as a heatmap using an in-house script.

The NJ method was applied to construct a tree with the Nei genetic distance [19] using the read.PLINK and dist.genpop function respectively of ‘ape’ and ‘dartR’ package [20,21] in the software R v. 4.1.2 [22] and plotted with the Splitstree software [23]. Again, with the ‘ape’ R package [24] individual Neighbor-joining tree of distances was calculated using the dist.gene function relatively to Colombian breeds only, which was visualised by the ggtree R package [25].

In addition, a run of homozygosity (ROH) was estimated for each animal using PLINK and the following criteria were used: the minimum length was set to 1 Mb (--homozyg-kb), one missing SNPs and up to one possible heterozygous genotype were allowed in the ROH (--homozyg-window-missing 1 and --homozyg-window-het 1), the minimum number of consecutive SNPs included in an ROH was set to 30 (--homozyg-snp 30), the minimum SNP density per ROH was set to one SNP every 100 kb (--homozyg-density 100), and the maximum gap between consecutive homozygous SNPs was 500 kb (--homozyg-gap 500). The inbreeding coefficient based on ROH (FROH) for each animal was calculated as follows: FROH = LROH/LAUT where LROH is the total length of all ROH in the genome of an individual and LAUT is the specified length of the autosomal genome covered by SNPs on the chip (2452.06 Mb).

#### 2.2.2. Population Structure

The pairwise genetic relationships were estimated using a matrix of genome-wide identity-by-state (IBS) genetic distances calculated by PLINK [15]. After, the results were plotted by multidimensional scaling (MDS) using the first three dimensions as implemented in the R package ‘plotly*’* [26], to explore the genetic differentiation of the whole dataset. With the same above methodology, a plot of MDS was produced for the Colombian breeds only and graphed using the first two dimensions thanks to the ggplot2 package of R [27].

After a quality control filter, LD pruning was performed on the quality filtered data (both for the global and for Colombian breed-only datasets) using the “--indep *50 2 0.1*” PLINK [15] command (25385 and 12446 SNPs remained for this analysis respectively). Additionally, the representative.sample function into the BITE R package [28] was used in the datasets to force to 30 and 20 individuals in global and Colombian datasets, respectively, each breed sample size using Identity-by-State matrix (IBS) and principal components data. On this new set, the Admixture software v1.3 [29], which uses a model-based clustering algorithm, was used to determine the optimal number of k clusters (i.e., the most likely number of contributing populations) and describe individual ancestry in these clusters. Using the “—*cv*” flag, a 20-fold cross-validation procedure was performed for a range of *k* (between 2 and 50 for the global set and 2 to 15 for the Colombian breeds data set). Cross-validation error per K plots are reported in Appendix A for each dataset. Cluster assignments for the global dataset, ranging from k = 2 to k = 10, were graphically presented using the Q matrix of membership coefficients with the membercoef.circos function of the R package ‘BITE’ [28], while for the Colombian dataset, k = 2 to 5 and 10 were represented using the ‘ggplot2′ package [27] in R.

A NetView analysis using the R version [30], which uses an unsupervised clustering method known as Super Paramagnetic Clustering (Spc), was applied to detect fine-scale population structures based on genetic distances. The program requires a specification of the maximum number of nearest neighbours (K-NN) an individual can have. K-NN was set to 10 and 30 to investigate the characteristics of both fine- and large-scale genetic structures after determining the most likely community starting point using the “*plotSelection”* option (Appendix A). Following clustering, population networks were visualised using a different colour for each breed.

### 2.3. Haplotype Sharing

Haplotype sharing was estimated using the complete marker set after quality filters, as determined by identity-by-descent estimations among individuals. The data file was first phased, and missing genotypes were completed by imputation using Beagle v4.1 software [31,32]. Haplotypes were reconstructed using the Refine IBD software [33], setting 40-SNP sliding windows, a minimum haplotype cM length of 1.5, trimming of 0.15 cM, and LOD score of 3. Results on the number of haplotypes shared between different breeds (shared within the same breed set to 0) were plotted using the ‘chordDiagram’ function of the R package ‘circlize’ [34].

## 3. Results

### 3.1. Analysis of Whole-Genome Diversity

The highest H_e_ values were found in OPC_E_ and OPC_S_ (H_e_: 0.39), while for OPCP, and OPCW, values between 0.36 and 0.35 were found. Similar results have been observed for BCL and NCL sheep breeds (Table 2). Considering an average value of 0.32 [16], these results show that Colombian Creole hair sheep have higher genetic variability than other international sheep breeds. On the other hand, the observed heterozygosity (H_o_) varied between 0.37 in OPC_P_. At the same time, it was OPC_S_ for 0.36 and 0.35 for OPC_E_, and for wool sheep (NCL and BCL) indicated values between 0.31 and 0.35, respectively. These results show that the (Ho) values were consistently lower than (He), except for OPCP. Average MAF values (Table 2) were uniform among the breeds considered, ranging between 0.28 (OPCE, OPCS, OPCW), 0.26 (OPCP, BCL), and 0.27 (NCL). The Hardy–Weinberg equilibrium deviation (*p* value in Table 2) was not significant in Colombian breeds. OPCE and OPCP showed negative values (−0.06 and −0.04) for F_is_ (Table 2), indicating an excess of heterozygosity in these populations, while positive values were identified for OPCS, OPCW, NCL, and BCL, suggesting a certain degree of inbreeding.

The ROH coefficients (Figure 1) of the sheep breeds studied were Low with a mean (in all breeds) of 0.07, with the BCL having the highest value (0.094), in contrast to the NCL, with an ROH value corresponding to 0.045. While OPCP, OPCS and OPCW hair sheep show similar values (0.06), while OPCE indicated the highest value (0.085).

Pairwise F_ST_ results (Figure 2), calculated on the whole dataset, were low among Colombian breeds but sufficient to differentiate between breeds. The average FST value among the six populations was 0.05. Low values of F_ST_ were found between the OPCP with the other Caribbean hair breeds and the wool criollos from Colombia (FST~0.06 and 0.09), while the hair Creoles from the Caribbean (OPCE, OPCS, OPCW) and wool Creoles (BCL and NCL) indicated a low value of F_ST_ (0.01 and 0.05). At a global level, the OPC_P_ showed low values of FST with all Brazilian hair breeds (0.11) and Barbados Black Belly from the Caribbean (0.08). The same happens with OPCE, OPCS, OPCW, and the wool BCL. The NCL breed generally showed low FST values compared with other wool-type American breeds, and the closest relationship was with the Brazilian Wool Creole (BSC) (0.05). WAD, from the Atlantic coast of Africa, showed a lower value with OCPE, OCPS, and OCPW (<0.1). Creole hair sheep from Colombia showed a low F_ST_ (<0.16) with some African breeds such as the African White Dorper (AWD), African Dorper (ADPR), Egyptian Barki (EBI), Norduz (NDZ), Qezel (QEZ), Ronderib Afrikaner (RAD), and Ethiopian Menz (EMZ). In contrast, MacArthur Merino (MCM) and Namaqua Afrikaner (NQA) breeds show an F_ST_ > 0.27 with all the Colombian Creole breeds.

### 3.2. Genetic Relationship and Admixture

In the MDS analysis (Figure 3), as well as the six breeds from Colombia, an Ecuadorian hair-type breed (ECU, Table 1), an additional 38 breeds from different world countries have been included for comparison (Appendix A). The selection criteria were based only on those breeds that could have contributed to the development of the Colombian Creoles, based on historical documentation: Iberian breeds as representatives of colonial age introductions, African breeds to infer introgressions of late colonial age or more recent introduction [0], and finally, international and exotic breeds such as the African Dorper, or Australian Merino types to infer the influence of recent crossing [35]. The MDS with 44 breeds showed a genetic relationship between the Colombian Creole Wool sheep (NCL, BCL) with the Iberian sheep (CAS, CHU, OJA and RRAA) and other European breeds (MIKL, MLA and SBS). WAD was related to the African cluster but was near to Colombian hair sheep breeds. In contrast, the Creole hair types (OPCE, OPCS, OPCP, OPCW) and sheep from Ecuador (ECU) and some isolated individuals of the Colombian Creole wool-type (BCL) grouped jointly and are genetically related to Barbados Black Belly (BBB). The Brazilian hair sheep (BNG, SINES, MNOVA, FATT, BSOM) formed independent groups. In addition, there was a clear separation of the American Creole hair types from the African populations (EMZ, NAQ, RAD, RMA) and the rest of the European, Middle Eastern, Oceanian and Asian breeds. BNG was in an intermediary position between Near East, European and Caribbean sheep breeds.

A Neighbor-Net network constructed using 42,294 SNPs generally grouped all the populations according to their geographic origin (Figure 4). Colombian hair types (OPCE, OPCS, OPCW) were genetically close despite phenotypic differences and geographical distribution. The OPCP was also closely related to the BBB and WAD as expected since they are phenotypically similar and share a common origin and history. Although they are in the same branch, the Brazilian Creoles (BNG, FATT, MNOVA, SINES) formed an independent cluster. Wool Creole sheep breeds, NCL and BCL, are grouped closely with the Ecuadorian Creole (ECU). The Brazilian Somali (BSOM) shows close genetic relations to the African RMA, EMZ, and other African breeds such as AWD, ADPR RAD, and NQA. The Saint Elizabeth sheep (STE) was in an intermediate position. While Iberian, European, Asian, and Middle Eastern sheep form independent groups, these results were consistent with the MDS analysis (Figure 3).

The Admixture in Figure 5 shows the K2 to 10 representing the most crucial clustering pattern of the global admixture hypothesis. The cv plot (Appendix A) showed a plateau starting at K = 30 and reaching the lower value at K = 38. The results in addition to reconfirming the result observed in the previous statistical analyses, we can observe a clear separation of the African races (blue colour). The European races, jointly with some of the Caribbean areas (e.g., GCN and STE) and South American (BCS), are grouped from K2 to K4 (red colour). The European component identified by the red colour remains visible up to k3 or K4 in the Middle East breeds (EBI, QEZ, IAW) and the Egyptian EBI. Interestingly, this European influence remains visible at k2 and 3 in OPCP, OPCE, OPCS, and OPCW but much clearer in NCL and BCL. OPCP separates itself starting from k9 from the other Colombians, as has already been reported in the network results. NCL and BCL show an evident influence of the k = 4 component of the European breeds and remain well differentiated from the other Colombian hair sheep breeds. Some individuals, however, in particular in BCL, show a specific component in common with the Colombian hair sheep breeds, with the BBB and ECU to testify to the presence of a certain introgression between them.

The haplotype exchange analysis between populations (Figure 6) is based on the haplotype count matrix where flows within the same population and flows count less than 50 were set to 0. It allowed us to identify a more pronounced haplotype exchange between OPCPS with other hair breeds; Barbados Black Belly (BBB), Brazilian Black Belly (BNG), Ecuador (ECU), and Gulf Coast Native (GCN), but also with the BCL wool creoles and to a lesser extent with the NCL. Haplotype exchange with African breeds exists. For example, Colombian hair-type (OPCE, OPCP, OPCS, and OPCW) share haplotypes with EMZ and NQA and, as expected (is a common breed reared in the American tropical environment) with AWD.

These results align with what was found in the ADMIXTURE analysis (Figure 5) when it is assumed that there are 20 ancestral populations (K = 20). Creole hair sheep from the Colombian Caribbean, for example, the OPCW, show a significant exchange of haplotypes with Colombian wool sheep (NCL, BCL). Although they are geographically close, OPCW does not exchange haplotypes with OPCE and OPCS. At a global level, it was found that OPCW shares haplotypes with BNG, BSOM, and ECU. The OPCS does not share haplotypes with OPCE, while the BCL and NCL wool Creoles show a significant exchange of haplotypes. They share haplotypes with BBB, BNG, BSOM, ECU, and GCN, with the same tendency as OPCE. The NCL shares haplotypes with the BCS, GCN, and ARA when looking at the wool creoles.

### 3.3. Within Colombian Breeds Genetic Structure

Figure 7 shows the phylogenetic tree reconstructed from the analysis of 237 Creole sheep from Colombia. The dendrogram based on the individual phylogenetic reconstruction shows a separation of OPCP sheep from those of the Caribbean region (OPCE, OPCS, OPCW) and the wool Creoles (NCL, BCL). In a second grouping, the Colombian wool Creole and hair-type sheep are grouped, despite their geographical distribution and the difference in their phenotypes. This may be since they have a common origin, and there is little differentiation by region despite the difference in the climate.

The MDS graph constructed using the prime due dimensions for the sole six Colombian populations (Figure 8) shows a significant separation between the OPCP population and the other correspondents to the Caribbean and Andean region (OPCE, OPCS, OPCW and BCL, and NCL). In the upper left part of the graph, there appears to be a cluster format of OPCE, OPCW, and BCL with numerous individuals apart from OPCS. The remaining part of the OPCS individual with the NCL population is situated in the lower left part of the graph.

The population structure for the six Colombian sheep breed sheep was inferred separately by ADMIXTURE (Figure 9), considering a range of 2 to 5 and 10 potential clusters (K). The most likely K to explain the population’s genetic structure is K = 3 using the coefficient of validation (Appendix A). At K = 2, OPCP presents a genomic component that is different from the other breeds, while the hair-type creoles (OPCW, OPCS, OPCE) and the wool-type Creoles (BCL, NCL) share the same genomic information. When it is assumed that there are three ancestral populations (K = 3), a common component can be found for all sheep breeds, consistent with the mixed origin of these populations, except for the OPCP. At K = 4, the OPC_E_ differentiated from the OPCW, OPCS, BCL, and NCL. At the same time, the OPCP remains heterogenous but differentiated from the rest across all the tested K. At a higher value of K, the populations were progressively assigned to separate groups but with no homogeneous classification of every single individual. NCL and BCL breeds have a common genomic background, even if the NCL has a lower degree of admixture than the BCL. Concerning the hair-type creole sheep, both a high degree of admixture and an intense subdivision into sub-groups within each breed entity are appreciable at K 10 and 14. Curiously, although the OPCW is geographically separate and phenotypically different from the wool creole, they share a common genomic component with NCL and BCL, confirming the previous analysis.

Figure 10 presents network graphics for K-NN = 10 and K-NN = 30. At a K-NN of 10, network interconnections between most OPCS and OPCE individuals (pink and blue circles) with OPCW samples appeared. Other samples from OPCE and OPCS formed independent groups isolated by the central cluster. BCL and NCL clustered jointly as a separate group even if, as also evident in MDS analysis, some BCL individuals shared a connection with OPCS. OPCP formed two homogeneous groups separated by the other samples. This K-NN value resulted in the formation of clusters and networks mainly based on the geographical origin of samples.

Using a K-NN of 30, all samples showed significant interconnection in a cluster where OPCP is at one extreme, connected to a central group constituted by OPCS, OPCW, and OPCE. NCL and BCL formed a divergent group directly connected to OPCE and OPCW individuals at the other extreme. Interestingly, some OPCE and OPCS samples remained isolated in independent groups even at this level.

## 4. Discussion

South American sheep breeds, called creoles, are typically animal descendants from the introductions in the colonial age. Still, this term is generally used to define animals that are in a certain way autochthonous of the South American continent.

For decades, Colombian creole hair sheep (OPCE, OPCS) have been misnamed by foreign names, while the scientific community has erroneously denoted them as Ethiopian Sudan [5]. Meanwhile, Revelo et al. [2], found no clear genetic relationship with African genetics type sheep using mitochondrial DNA data. In addition, the genetic makeup and origin are still unclear, although approaches have been made using microsatellites [5,36], mtDNA molecular markers [2] and, recently, SNPs [13]. We present the first genomic results for the OPCE, OPCP, OPCS, and the Wayuu breeds (OPCW), a genetic resource reared by the homonymous indigenous peoples, constituting an invaluable resource for their food security. These people live in the desert of La Guajira with extreme temperatures and low-quality pasture [2]. This study also presented the first genomic analysis for wool-type Creole sheep breeds from Colombia; the NCL sheep in the Nariñense plateau and the BCL in the Colombian Andean region.

The diversity parameters of the six breeds distributed throughout the national territory (Table 2) indicated high He values (for example, 0.31, 0.37). Ortiz et al. [13] reported similar values for Creole hair sheep from the inter-Andean valleys of Colombia (He = 0.357 and He = 0.396). Paim et al. [3] found lower He values (0.29 and 0.35) in five Brazilian hair-type sheep breeds than in this study. In the same way, Grasso et al. [37] reported lower values in wool sheep from Uruguay (Ho 0.28 and Ho 0.25). The Colombian creole hair sheep have greater genetic diversity than the Brazilian hair types. The lower variation in Brazilian breeds may be due to their participation in breeding programs (Santa Ines and Morada Nova), crossbreeding (Santa Ines), closed flocks (Somali), or low population size (Brazilian Black Belly and Fat Tail).

On the other hand, Spangler et al. [1] in Creole hair sheep from the Caribbean reported H_e_ values (0.30 and 0.35) lower than those found in Colombian Creoles. Comparing Colombian Creole sheep with African breeds, for example, the Namaqua Afrikaner breeds (H_e_ = 0.22) and other European breeds such as the Merino (H_e_ = 0.321) [38], H_e_ was lower than those found in this study. This difference seems to be related to the recent crossbreeding experienced by Colombian breeds. Although European breeds show higher heterozygosity values, Deniskova et al. [39], using the same number of SNPs in 25 sheep breeds in Russia, found the highest H_e_ for the Baikal Fine-fleeced breed (0.39). Similarly, Mastrangelo et al. [40] reported that H_e_ values for Sicilian breeds range between 0.37 and 0.39.

The proportion of the autosome covered in ROH varied both within and across breeds. The NCL breed tended to have fewer ROH, whereas large inter-animal variability existed within the BCL, OPCE, and OPCS. Purfield et al. [41] and Mastrangelo et al. [42] described lower values in commercial and local Italian sheep breeds, respectively of 0.025–0.319; and 0.099–0.016. A lower value and a sharper distribution of ROH data could be due to a smaller effective number and geographical isolation that is compatible with the actual situation of NCL and OPCW. These results could be useful in the future for the management of inbreeding and to prioritize conservation actions. The higher estimates of diversity in the Creole sheep of Colombia compared to the breeds of America are probably related to a diverse origin, started during the colonial period [3,8,43,44,45] and continued in the subsequent introduction of animals originated in West Africa [1,2,5,36]. Evolutionary forces such as natural and artificial selection, genetic drift, and crossbreeding with commercial breeds contributed to these animals’ genetic diversity. On the other hand, positive F_IS_ values were found for OPCS, OPCW, NCL, and BCL (Table 2). Palacios [46] documented that in most of these production systems in the Caribbean region, producers do not select animals, and crossbreeding is performed randomly; this same situation occurs with wool sheep in the Andean region (BCL), as well as in the Colombian Southwest (NCL). Ortiz et al. [13] indicated that the loss of heterozygotes and subsequent Hardy-Weinberg disequilibrium in subpopulations might be due to the Wahlund effect [47]. This effect could be compatible with the geographical isolation of several flocks included in this study, confirmed by Admixture and Netview analysis. In contrast, OPCP presented a low negative value (−0.04, Table 2). The presence of geographical isolation and divergent groups of animals brings together an advantage and a disadvantage at the same time: on the one hand, it is an indication of genetic variability and local adaptative dynamics, but, on the other, it highlights the need for conservation policies to prevent these isolated groups from being lost [48]. Although these populations retain a high genetic variability and the actual number is still acceptable need protection and conservation, especially from the risk of disease outbreaks. The original study provided a new frame of information, and data, necessary to support future policy creation and management plans.

In the Valle del Cauca nature reserve, where these animals were sampled, directed matings and a certain degree of management and selection are carried out, including ram interchange, which would probably explain the small inbreeding in these populations. Vivas et al. [5] indicated that when there is high genetic diversity and the absence of intrapopulation inbreeding, it is attributed to the constant crossings of Creole sheep with commercial breeds, as this seems to be happening with the OPCE (−0.05) and the Creoles of Ecuador (ECU −0.06). Paim et al. [3] found positive F_IS_ values (0.084 and 0.224) in seven Brazilian sheep breeds (five hair types and two wool types), slightly higher values than the OPCS, OPCW, NCL, and BCL breeds.

The population structure for OPCP was the most divergent among the Colombian populations, observed in all the statistical tests. Caicedo [49] reported that in the El Hatico Nature Reserve (RNH), in 2005, few Pelibuey sheep were imported from Mexico to improve the extant creole populations to increase productivity. Although the process was incomplete, this could explain the divergency of OPCP from the OPCE, OPCS, and OPCW, confirmed in the Netview at K-NN = 10, where two homogeneous groups separate OPCP from the other breeds can be observed (Figure 10).

On the other hand, the Colombian hair types OPCE, OPCS, and OPCW, although phenotypically different [2,50], seem to have a common origin. According to historical documents and scientific reports, the Canary Islands played a decisive role in the distribution of animal genetic resources after the discovery of America by becoming a crossroads for Spanish navigators and other European countries with their overseas colonies. There is considerable historical evidence confirming the Canarian sheep’s participation in the first colonization of America [51,52]. In the Canary Islands, an autochthonous hair sheep of African origin exists (the only of this type in Europe), and, probably, It could have been imported to the Americas during colonial times. Delgado et al. [43] highlight Canarian hair sheep’s influence in forming the genetic base of Caribbean sheep and their subsequent spread to the rest of the continent.

However, the arrival of the first hair sheep in the country is unclear. Probably some specimens arrived at the time of the slave trade from the African continent [1,2]. The possible entry route was the Atlantic coast used by merchants from the Magdalena department who traded with Aruba and Curaça and smugglers who travelled between the Caribbean islands and Guajira [53]. This historical information probably explains the genetic closeness of OPCS and OPCW, despite the geographical isolation between these groups. However, no shared haplotypes were found between OPCE and OPCS (Figure 6). This same trend is shown in Admixture when a K = 14 up to K = 20 is assumed. In the Netview analysis, both at K-NN = 10 and K-NN = 30, the majority of the OPCE individuals form independent groups, in agreement with Vivas et al. and Florez et al. [5,14].

Another important finding was that wool (NCL, BCL) and hair types (OPCE, OPCS, OPCW), although geographically separated, presented very little genetic differentiation (F_ST_ < 0.014). Similar results were obtained by Revelo et al. [2] with mtDNA markers and by other works with STRs markers [5]. However, on the other hand, groups of animals identifiable with some sampling regions are quite divergent, as in Netview results. Geographic-based divergency is compatible with the hypothesis of ancient and isolated (1600–1700?) introgressions of African genetic material, through the passage from the Caribbean islands or directly with the slave trade, on an already existing substrate of animals of Spanish origin, introduced around 1493–1500 during the entry of the Spanish colonizers [1].

However, at least two small introductions of animals directly from Africa are known. In 1940, across the Atlantic coast, a group of red Ethiopian sheep arrived from what was known as Abyssinia (Ethiopia), while in 1970, 25 Persian breed specimens were introduced to the Huila region near Cali [53]. These animals spread into the areas of Armero, Honda*,* and Venadillo through crossbreeding. However, today it is difficult to know what the pre-existing bases were in the country due to the lack of historical records. Today, depending on the area, Colombian people refer to the Ethiopian or Sudan to distinguish hair-type animals from wool-type ones precisely in reminiscence of these importations. In many cases, other names such as “oveja Africana” (African sheep) or camuro (probably from the proto-Italic kameros, “to bend, curve,” referring to the hair structure) are used. In the global comparison, both the DMS (Figure 3) and Admixture (Figure 4) analyses indicated that the OPCP shows a close genetic relationship with the Barbados Black Belly (BBB), as previously commented. A common origin and history, as supported by the number of haplotypes shared between breeds can be supposed. The OPCP showed, in addition to sharing haplotypes with BBB, a close genetic link with BNG, BSOM from Brazil, and ECU from Ecuador. Probably this relatedness is a consequence of the fact that some Brazilian hair sheep first entered Venezuela and Colombia at the time of the colony [43,51].

The WAD breed deserves an additional discussion as, being a West African genetic type, it would represent the possible introductions of animals during the late colonial era. Spangler [1] proved some relationship with the WAD and other breeds of the Caribbean, and in our results, it clustered very near to Colombian hair-type within BBB. In our work, we do not find many haplotype sharing interchanges of WAD with Colombian breeds as between Colombian hair types and Caribbean breeds. These results showed that the introduction process in Colombia does not happen directly but gradually and through the Caribbean islands, used as a hub for the diffusion of domestic animals into the Americas as described for cattle in other research [54]. this result is in line with what has been reported in the bovine species also in historical documents [52] and could be the basis for continuing new studies on the diffusion scenarios of livestock in the continent.

In all the statistics, genetic divisions were detected that separate the European, African and Brazilian sheep from the Creole sheep of Colombia. Similar results were reported by Paim et al. [3], Kijas et al. [38], and Spangler et al. [1]. They reported a clear genetic division of sheep from the Americas (Brazil and the Caribbean) from those European, African, or Asian populations. These authors concluded that the Caribbean breeds’ genetic origin is common with African animals mixed with those of Mediterranean Europe. NCL and BCL, in the Neighbor network constructed from Nei’s genetic distances, are grouped with Ecuadorian hair sheep (ECU). These results are consistent with the Admixture plot in all K and the MDS (Figure 3 and Figure 5). The genetic link between these breeds is consistent with that Colombia, Ecuador, and Venezuela represented a single national entity (La Gran Colombia), separated into three nations only in the late 1830s. Hence, the history of the animal genetic resources and the three countries is strongly interconnected.

## 5. Conclusions

The extant scientific information on south American sheep breeds is limited because they are usually reared in marginal and low input systems in a region where cattle is the most important genetics resource. However, sheep are strategically important for many people in South America for their adaptive capabilities to the different agroclimatic conditions. Promoting their recognition and characterizing their production ability and genetic uniqueness is necessary. In this scenario, the knowledge of the population structure and the genetic diversity of the Colombian Creole sheep becomes essential. In this work, we analyzed all sheep breeds in Colombia, including the Wayuu breed (OPCW). The results show that the genetic basis of Colombian sheep is of European origin, as seen with mitochondrial DNA. Subsequently, various introductions of animals of African origin followed through different geographical routes and at different historical moments.

## Figures and Tables

**Figure 1 genes-13-01415-f001:**
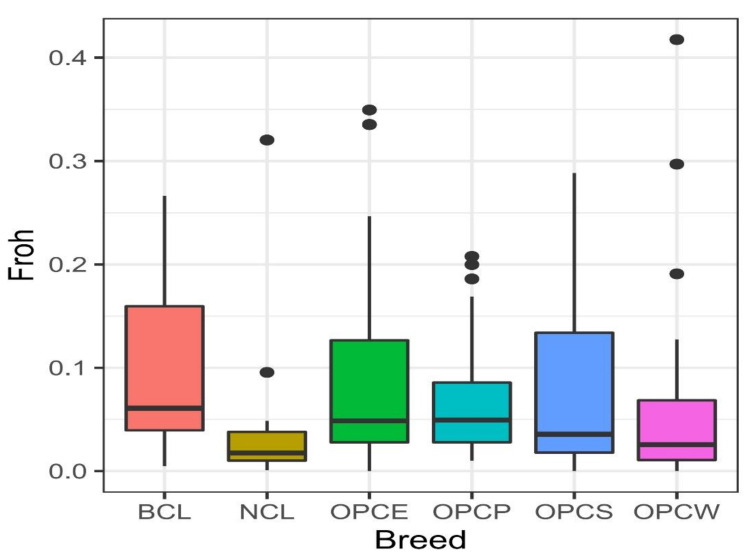
Distribution of runs of homozygosity inbreeding coefficients for each Colombian population.

**Figure 2 genes-13-01415-f002:**
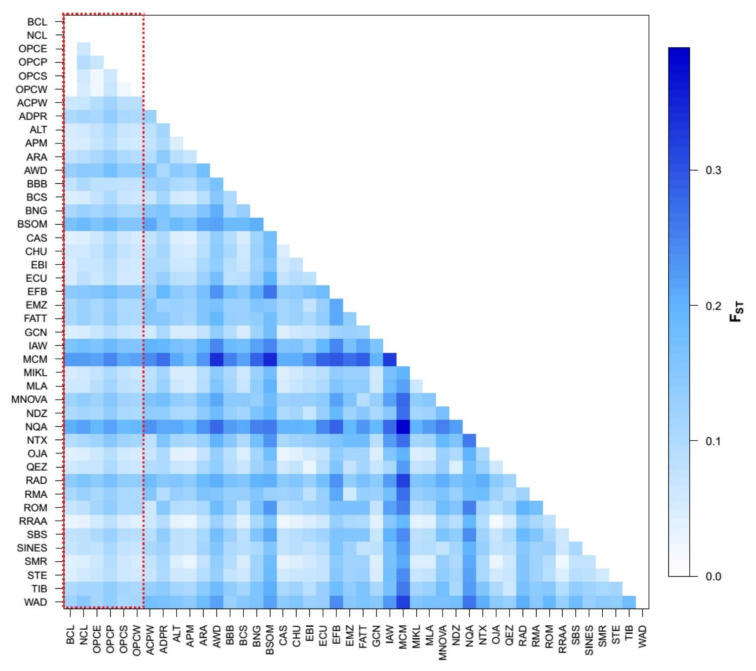
Genetic differentiation pairwise FST Matrix values among 44 sheep populations/breed. the dashed red line highlights the Colombian breeds.

**Figure 3 genes-13-01415-f003:**
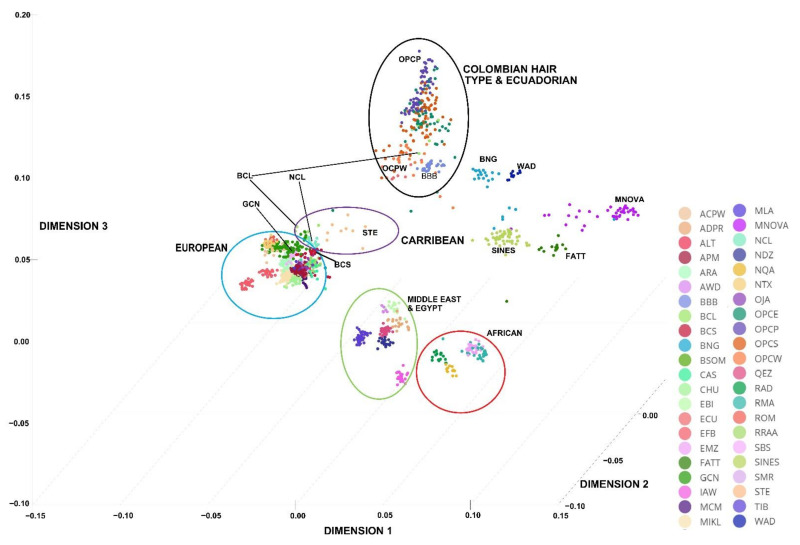
3D MDS plot showing the genetic relationships among all sheep breeds analysed. For a full definition of the breeds, see Appendix A. The red circle defines the cluster of African breeds, green those of the Middle East including Egypt, the light blue that of the European breeds including Oceania, purple that of the Caribbean breeds and finally black that of the Colombian hair-type breeds.

**Figure 4 genes-13-01415-f004:**
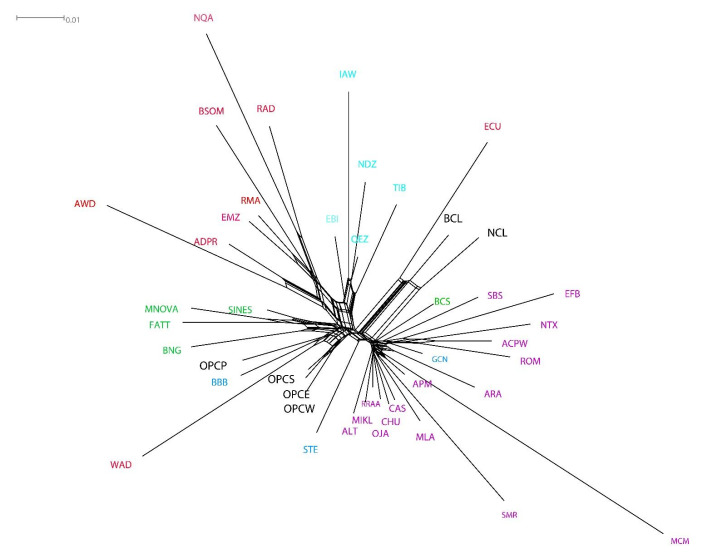
Neighbor network constructed from Nei’ genetic distances. In black the Colombian sheep breeds. Origin of breeds are indicated by colours: light blue, Asian, Middle East including Egypt; red, Africa; green, South-American native; blue, Caribbean; violet, European including Oceania.

**Figure 5 genes-13-01415-f005:**
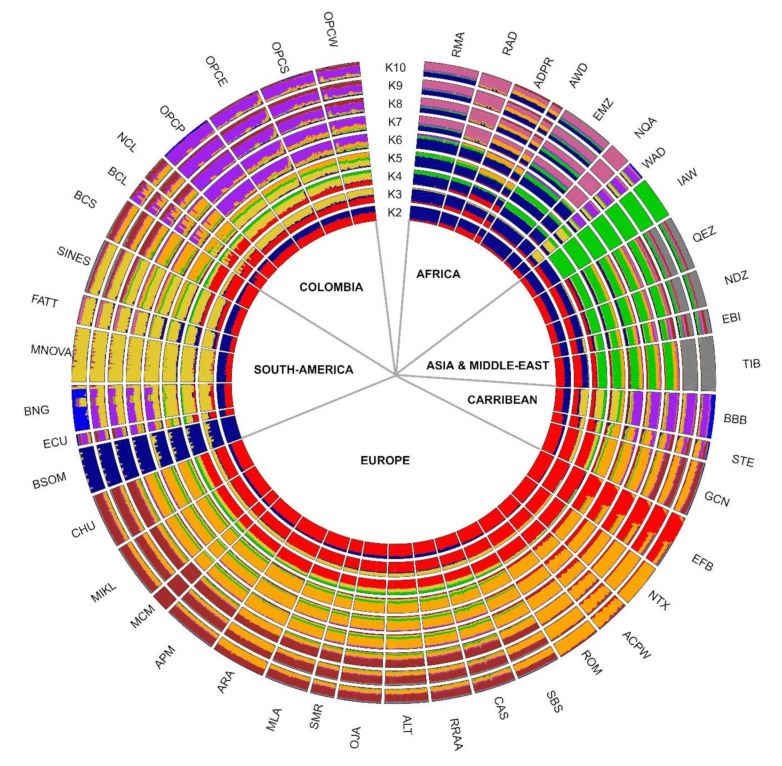
Clustering of 44 sheep breeds based on Admixture estimation for K = 2 to K = 10. Each bar indicated an individual and each colour the membership to a particular cluster; the number of clusters is determined by the K value. The sector inside the graphic indicated the geographical precedence of the breed: Africa, Asia and Middle-East including Egypt, the Caribbean, Europe including Oceania; South America and Colombia.

**Figure 6 genes-13-01415-f006:**
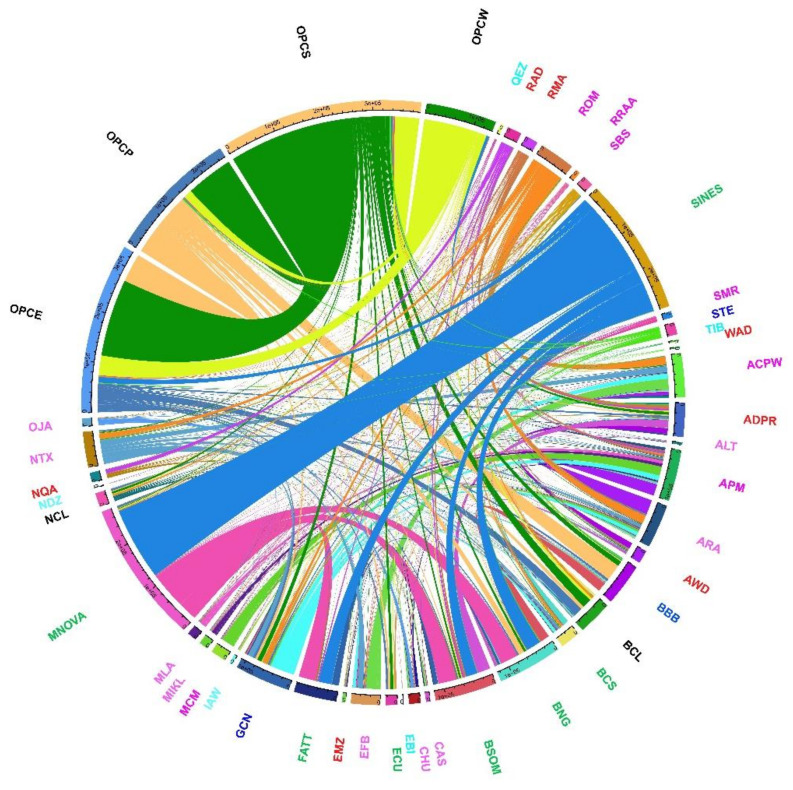
Circos plot of the number of haplotypes shared between breeds. Each coloured band. each coloured band represents the total number of haplotypes that are divided between a pair of races, where a greater thickness of the band represents a greater number of haplotypes. The flows within the same populations and those where the count is less than 50 are excluded from the graph. In bold are the Colombian sheep breeds. Origin of breeds are indicated by colours: light blue, Asian, Middle-East including Egypt; red, Africa; green, South-American native; blue, Caribbean; violet, European including Oceania.

**Figure 7 genes-13-01415-f007:**
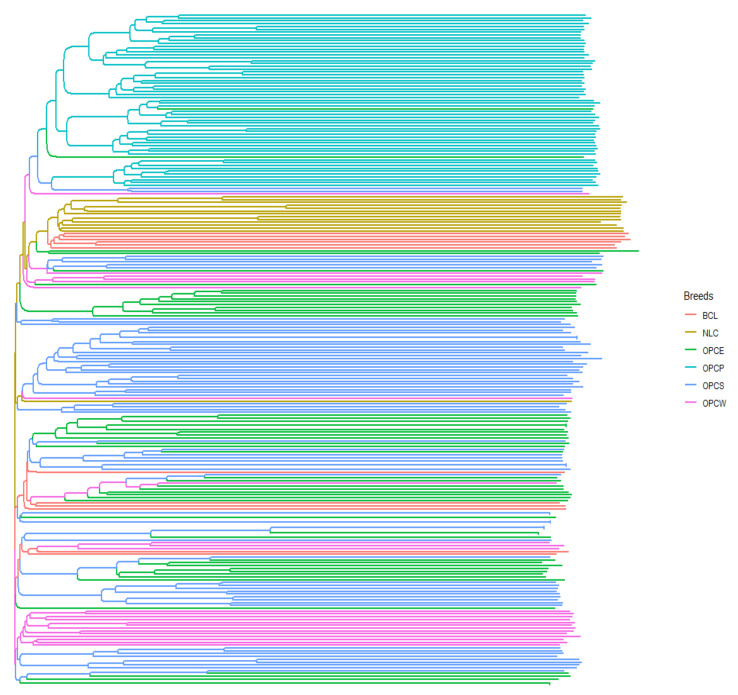
Dendrogram of individual pairwise genetic distances.

**Figure 8 genes-13-01415-f008:**
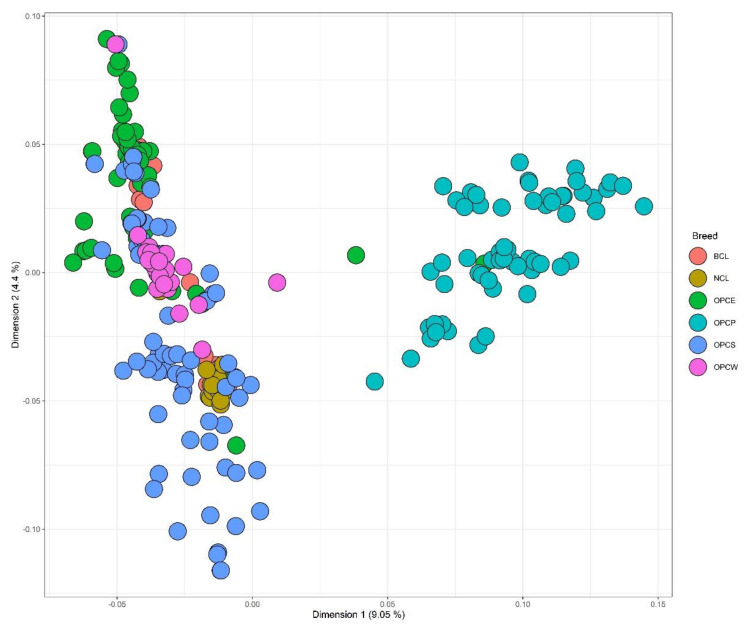
MDS plot constructed using the first two dimensions, showing the genetic relationships among the six Colombian populations (OPCP, OPCE, OPCS, OPCW, NCL, BCL).

**Figure 9 genes-13-01415-f009:**
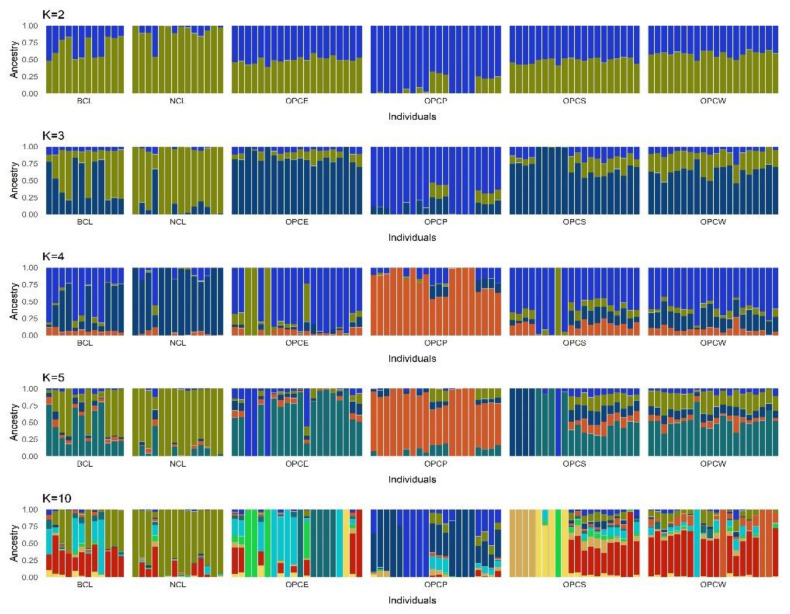
Model-based clustering using genotypic data from seven sheep breeds. K, Number of clusters, from 2 to 5, and 10. The best prediction model was K = 3. Each bar indicated an individual and each colour the membership to a particular cluster, the number of clusters is determined by the K value.

**Figure 10 genes-13-01415-f010:**
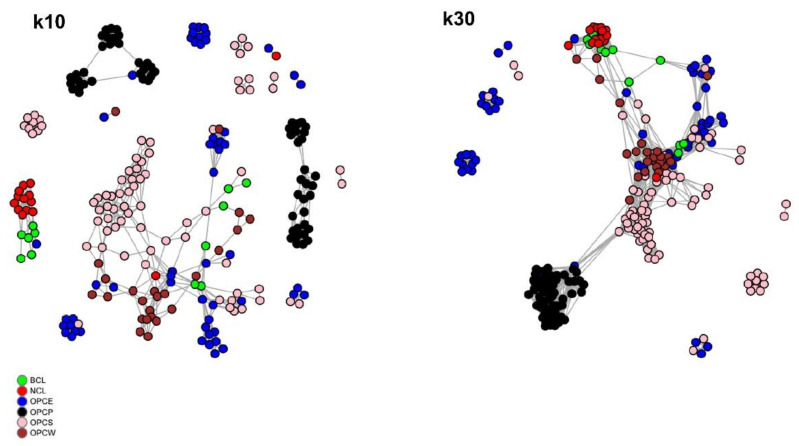
Visualisation of high-resolution population networks for 6 Creole sheep breeds. K-NN = 10 (left) investigated small-scale structures, K-NN = 30 (right) targeted large-scale structures and breeds are distinguished by node colour.

**Table 1 genes-13-01415-t001:** List of Colombian Creole sheep breeds, typologies, and geographical positions.

Breed	Acronym	Type	N(h)	Department (Region)
Ethiopian	OPCE	Hair	54 (6)	Córdoba, Cesar, Atlántico, Magdalena (Caribbean coast)
Sudan	OPCS	Hair	74 (5)	Córdoba, Cesar, Atlántico (Caribbean coast)
Pelibuey	OPCP	Hair	59 (3)	Valle del Cauca (Pacific Coast, Colombian southwest)
Wayúu	OPCW	Hair	24 (7)	Guajira Peninsula (Caribbean coast)
Wool Creole	NCL	Wool	12 (2)	Nariño (Andean region, Colombian southwest)
Wool Creole	BCL	Wool	14 (3)	Boyacá; (Andean region, Colombian southwest)

N: number of sampled animals; h: number of flocks considered.

**Table 2 genes-13-01415-t002:** Diversity indices in the six Colombian sheep breeds.

Acronym	Breed Name	Type	n	Ho	He	MAF	*P*	F_is_
OPCE	Etiope	Hair	54	0.35	0.37	0.28	ns	−0.06
OPCS	Sudan	Hair	74	0.36	0.37	0.28	ns	0.03
OPCP	Pelibuey	Hair	59	0.37	0.35	0.26	ns	−0.04
OPCW	Wayuu	Hair	24	0.35	0.36	0.28	ns	0.05
BCL	Wool Creole Boyaca	Wool	14	0.31	0.35	0.26	ns	0.11
NCL	Wool Creole Nariño	Wool	12	0.35	0.36	0.27	ns	0.01

N-number of animals; Ho, observed heterozygosity; He, expected heterozygosity, MAF, mean minor allele frequency; *P*, Hardy-Weinberg disequilibrium deviation *p*-value; F, inbreeding coefficient estimates; ns-not significant.

## Data Availability

Data of Colombian sheep breeds are available to corresponding author upon reasonable request. The dataset from Caribbean sheep breeds (Spangler et al., 2017 [1], doi: 10.15482/USDA.ADC/1351110 can be accessed via http://dx.doi.org/10.15482/USDA.ADC/1351110 accessed on 26 March 2021. The remainder of the genotype files used for purpose of comparison are available in the ISGC website (http://www.sheephapmap.org/ accessed on 26 March 2021).

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
