# Peer review of "New Insight into the Genome-Wide Diversity and Admixture of Six Colombian Sheep Populations"

_genes, 2022, doi:10.3390/genes13081415_

Round 1

Reviewer 1 Report

The manuscript provides a comprehensive approach to understanding the genetic diversity and admixture of Colombian sheep breeds. The authors also included the data from worldwide sheep breeds to extend the analyses and discussion. Overall, the methods are appropriate, and the authors provided explanations for the choice of method. My concern about the options to choose the LD pruning and the representative samples which might be unnecessary since the authors have a medium SNP chip and the sample size is not big.

The authors also need to pay attention to the writing and proofreading.

 Line 15-16: Add the sample size,

Line 19-21: The authors should extend one or two sentences for the results; they are very empty now.

Line 45-49: provides the supporting references.

Remove the extra space and double space in the manuscript.

Line 78: add a reference

Line 82: Move  “N: number  of sampled animals; h: number of flocks considered.”  To the table footnote.

Line 95: change The to the

Line 112-119: Which function did the authors use in each package and which are the parameters used, please provide if possible.

Line 126-128: Why did the authors perform LD pruning?

Line 164-165: Which level of He is considered as high genetic diversity?

Line 168: What did the authors mean by these results in "These results show that the (Ho) was lower than (He), except for OPCP."

Line 168: MAF: Did the authors mean the average of MAF?

Line 171-173: What is the implementation of the Fis? I believe the authors should consider it in terms of inbreeding.

Table 2: F or Fis

Figure 1 is not informative; the authors might highlight the Colombian breeds with different colors.

In Figures 4 and 5: The authors should add more explanations in the footnote.

Line 300: Did the authors mean Figure 8? (Fig MDS and Network)

Line 413: it is not clear for

Line 455: Should it be 5. Conclusion

The authors might remove the references in the conclusion if they are already discussed in the manuscript.

L

Author Response

RESPONSES TO REFEREE 1

Dear refere thank you for useful suggestions and correction and to take time to perform the reviewing in this journal. Follow the answer to each point suggested.

The manuscript provides a comprehensive approach to understanding the genetic diversity and admixture of Colombian sheep breeds. The authors also included the data from worldwide sheep breeds to extend the analyses and discussion. Overall, the methods are appropriate, and the authors provided explanations for the choice of method. My concern about the options to choose the LD pruning and the representative samples which might be unnecessary since the authors have a medium SNP chip and the sample size is not big.

# AUTHORS: Thank you very much for your comments. LD pruning is generally a practice in Admxiture analyzes and is a prerequisite suggested by the author of the software (admixture manual, page 5). However, we also tested the analysis with the original dataset without obtaining substantial changes in the results shown, apart from a faster estimate, and for these reasons we opted to keep the pruning as reported by numerous other authors. The choice of reducing the number of samples to a homogeneous number among the races has instead a substantial effect on the estimation. While it does not change in a profound way, the result greatly alters the estimate of the coefficient of variation (Cv) and therefore the identification of the optimal K. For example, in the global dataset using the dataset without this expedient we do not obtain any inflection points using the values of Cv up to a k of 100 while we obtain a value of 38 (supplementary file 1) which is much more plausible in relation to the number of populations studied. The effect of the sample size balance is described by several authors who have studied the behavior of similar software such as Toyama et al 2020 (10.1111 / 1755-0998.13234). By other hand, the representative.sampling function of BITE package guarantee that the informativeness of the sample is retained.

The authors also need to pay attention to the writing and proofreading.

# AUTHORS: thank you for your comment and sorry for that.

Line 15-16: Add the sample size,

# AUTHORS: done

Line 19-21: The authors should extend one or two sentences for the results; they are very empty now.

# AUTHORS: done

Line 45-49: provides the supporting references.

# AUTHORS: done

Remove the extra space and double space in the manuscript.

# AUTHORS: done

Line 78: add a reference

# AUTHORS: done

Line 82: Move  “N: number  of sampled animals; h: number of flocks considered.”  To the table footnote.

# AUTHORS: done

Line 95: change The to the

# AUTHORS: done

Line 112-119: Which function did the authors use in each package and which are the parameters used, please provide if possible.

# AUTHORS: done

Line 126-128: Why did the authors perform LD pruning?

# AUTHORS: thank you for the observation. A complete justification has been reported in the general comments above.

Line 164-165: Which level of He is considered as high genetic diversity?

# AUTHORS: thank you for the observation. Following KijasJW, et al. (2009). PLOS ONE 4(3): e4668. https://doi.org/10.1371/journal.pone.0004668 we considered an average expected heterozygosity as 0.32. We added a comment regarding this choice in the text.

Line 168: What did the authors mean by these results in "These results show that the (Ho) was lower than (He), except for OPCP."

# AUTHORS: thank you for the observation, we modify the sentences as follow in a clearer way ”These results show that the (Ho) values were always lower than (He), except for OPCP”

Line 168: MAF: Did the authors mean the average of MAF?

# AUTHORS: yes, we specify it in the text.

Line 171-173: What is the implementation of the Fis? I believe the authors should consider it in terms of inbreeding.

# AUTHORS: thank you for the comment, we specify in the text.

Table 2: F or Fis

# AUTHORS: corrected.

Figure 1 is not informative; the authors might highlight the Colombian breeds with different colors.

# AUTHORS: done

In Figures 4 and 5: The authors should add more explanations in the footnote.

# AUTHORS: done

Line 300: Did the authors mean Figure 8? (Fig MDS and Network)

# AUTHORS: yes it is figure 8, sorry for that and thank you for the comment.

Line 413: it is not clear for

# AUTHORS: thank you for the observation. The sentences have been deleted.

Line 455: Should it be 5. Conclusion

# AUTHORS: done

The authors might remove the references in the conclusion if they are already discussed in the manuscript.

# AUTHORS: done

Reviewer 2 Report

The manuscript is interesting and scientifically sounds. The authors attempt to untangle the complex relationship and ancestry of Creole sheep. However, there some points, which should be addressed before final decision on the manuscript.  

Comments

The Abstract should be revised thoroughly because this section is not informative and clear in present form. Please do use full names of breeds (not acronyms).

L15-17 sample size might be indicated here.

L17-18 the HapMap Project might be mentioned here.

L 18 -19 This phrase is vague and should be clarified (for example, which genetic parameters were estimated) or removed.

L 19-20 This may be supplemented by the Fs values.

L 20-22 «animals from Mexico» and «animals of African origin» - which animals? This will be unclear to the potential readers. The breeds might be mentioned here.  

L 37 please use Merino-derived

Please check the spelling for Barbados Black Belly breed through the text.

Figure 2. – MDS plot was missing.

Figures 3, 4 and 5 – The origin of the analyzed breeds should be highlighted on the figures. Please see examples in references (Kijas et al., 2012).

L 246 – fur breeds? Or hair?

The Institutional Review Board Statement is missing.

Author Response

RESPONSES TO REFERE 2

Dear refere thank you for useful correction and advice. We reported the answer to each point suggested below in red color

The manuscript is interesting and scientifically sounds. The authors attempt to untangle the complex relationship and ancestry of Creole sheep. However, there some points, which should be addressed before final decision on the manuscript.

Comments

The Abstract should be revised thoroughly because this section is not informative and clear in present form. Please do use full names of breeds (not acronyms).

# AUTHORS: done

L15-17 sample size might be indicated here.

# AUTHORS: done

L17-18 the HapMap Project might be mentioned here.

# AUTHORS: done

L 18 -19 This phrase is vague and should be clarified (for example, which genetic parameters were estimated) or removed.

# AUTHORS: thank you for the comment. We deleted the sentence.

L 19-20 This may be supplemented by the Fs values.

# AUTHORS: done, we also add the data in the Results section at line 196.

L 20-22 «animals from Mexico» and «animals of African origin» - which animals? This will be unclear to the potential readers. The breeds might be mentioned here.  

# AUTHORS: done

L 37 please use Merino-derived

# AUTHORS: done

Please check the spelling for Barbados Black Belly breed through the text.

# AUTHORS: done, thank you for the observation and sorry for the mistakes.

Figure 2. – MDS plot was missing.

# AUTHORS: sorry for that and thank you for the observation, the figure 2, present as separated file was deleted for mistake in the manuscript.

Figures 3, 4 and 5 – The origin of the analyzed breeds should be highlighted on the figures. Please see examples in references (Kijas et al., 2012).

# AUTHORS: thank you your observation and for giving the example. We have modified the pictures accordingly the suggestions.

L 246 – fur breeds? Or hair?

# AUTHORS: thank you for your observation. We rewrite the paragraph as follow for clarity: “NCL and BCL show an evident influence of the k = 4 component of the European breeds and remain well differentiated from the other Colombian hair sheep breeds. Some individuals, however, in particular in BCL, show a specific component in common with the Colombian hair sheep breeds, with the BBB and ECU to testify to the presence of a certain introgression between them.”

The Institutional Review Board Statement is missing.

# AUTHORS: added

Reviewer 3 Report

A brief summary (one short paragraph) outlining the aim of the paper and its main contributions.

The main goal of the study was to explore Columbian sheep populations and to assess genetic diversity in the sheep genetic resources. The paper provides a useful body of knowledge on sheep genetic resources of Columbia and is useful for the implementation of conservation and sustainable breed improvement strategies for the country.

Broad comments: (highlighting areas of strength and weakness. These comments should be specific enough for authors to be able to respond).

Generally, the paper is fairly written and a good read for which the authors should be congratulated. However, some general issues to highlight are below:

Abstract: Covers the key aspects of the study, research questions, methods, results and discussion. But there is need to align the study to practice/ application in real world.

Introduction:  Introduces fairly key literature consulted. However, there is need to clearly articulate the objectives of the study. The writing could also be improved to increase the readability.

Materials and method: Some minor correction and editing required

Results: Can be improved.

Discussion: Some minor typological errors. The discussion could still be enriched with additional breadth especially relating to practical application.

Specific comments: Refer to details of specific comments included in the text submitted.

Author Response

RESPONSES TO REFERE 3

Generally, the paper is fairly written and a good read for which the authors should be congratulated. However, some general issues to highlight are below:

Abstract: Covers the key aspects of the study, research questions, methods, results and discussion. But there is need to align the study to practice/ application in real world.

Introduction:  Introduces fairly key literature consulted. However, there is need to clearly articulate the objectives of the study. The writing could also be improved to increase the readability.

Materials and method: Some minor correction and editing required

Results: Can be improved.

Discussion: Some minor typological errors. The discussion could still be enriched with additional breadth especially relating to practical application.

Dear reviewer, we warmly thank you for taking the time to read and correct our manuscript and appreciate the compliments on the text. The comments and correction have been added in the text. Ad in particular:

  • some comment about application and implementation of results have been added in the discussion and conclusion part.
  • Text have been revised for syntax and minor spelling correction and to improve readability

Reviewer 4 Report

Nicely presented paper. Here are some suggestions to improve it:

* Formatting issues: Might be due to pdf conversion. MDS plot (line 216) is missing from pdf; numbering is not correct (lines 77,92,110 all have 1)

* Could you show PCA clustering of these breeds.

*Could you also add more analysis with genes, networks and pathways on these population with it effects on production or other traits. 

* Please perform the Run of Homozygosity (ROH) analysis and analyze genomic region of interests

Author Response

RESPONSES TO REFERE 4

* Formatting issues: Might be due to pdf conversion. MDS plot (line 216) is missing from pdf; numbering is not correct (lines 77,92,110 all have 1)

# AUTHORS: thank you for the comment. Yes was a formatting error, we added thew MDS again.

* Could you show PCA clustering of these breeds.

# AUTHORS: done, we have performed a MDS for Colombian breed only.

*Could you also add more analysis with genes, networks and pathways on these population with it effects on production or other traits. * Please perform the Run of Homozygosity (ROH) analysis and analyze genomic region of interests

# AUTHORS: Thank you for the comment and suggestions. We performed and analysis of ROH for the calculation of ROH based inbreeding coefficient. Regarding the other comment about selection signature, genes pathway and deep analysis of region of interest we had already imagined adding this type of results. However, the complexity of the population and the heterogeneity of the breeding area would lead to suppose different clustering strategies (based on phenotype? On altitude? On climate? On social environment?) and the use of different statistical methodologies for the study of region under selective influence or influenced by inbreeding. For this reason, also to not overload this manuscript, which we deliberately wanted to focus on diversity and population structure (these populations are mostly scientifically unknown in Colombia), we decided not to include this part, and, possibly reserve it for a second manuscript.

Round 2

Reviewer 4 Report

Thank you for addressing comments raised in the review.